# Information Needs of Next-Generation Forest Carbon Models: Opportunities for Remote Sensing Science

**Céline Boisvenue \*** and **Joanne C. White**

Canadian Forest Service (Pacific Forestry Centre), Natural Resources Canada, 506 West Burnside Road, Victoria, BC V8Z 1M5, Canada; joanne.white@canada.ca
\* Correspondence: celine.boisvenue@canada.ca; Tel.: +1-250-363-0600

**Abstract:** Forests are integral to the global carbon cycle, and as a result, the accurate estimation of forest structure, biomass, and carbon are key research priorities for remote sensing science. However, estimating and understanding forest carbon and its spatiotemporal variations requires diverse knowledge from multiple research domains, none of which currently offer a complete understanding of forest carbon dynamics. New large-area forest information products derived from remotely sensed data provide unprecedented spatial and temporal information about our forests, which is information that is currently underutilized in forest carbon models. Our goal in this communication is to articulate the information needs of next-generation forest carbon models in order to enable the remote sensing community to realize the best and most useful application of its science, and perhaps also inspire increased collaboration across these research fields. While remote sensing science currently provides important contributions to large-scale forest carbon models, more coordinated efforts to integrate remotely sensed data into carbon models can aid in alleviating some of the main limitations of these models; namely, low sample sizes and poor spatial representation of field data, incomplete population sampling (i.e., managed forests exclusively), and an inadequate understanding of the processes that influence forest carbon accumulation and fluxes across spatiotemporal scales. By articulating the information needs of next-generation forest carbon models, we hope to bridge the knowledge gap between remote sensing experts and forest carbon modelers, and enable advances in large-area forest carbon modeling that will ultimately improve estimates of carbon stocks and fluxes.

**Keywords:** forests; forest modeling; carbon

---

## 1. Introduction

Multiple research domains contribute to the scientific understanding of forest carbon accumulation and fluctuations, including: plant and tree physiology and genetics; plant, tree, and landscape dynamics; and atmospheric interactions between plants, landscape, and soil. Forests establish themselves through interacting with pedospheric and atmospheric conditions as well as other on-site vegetation. Trees and other vegetation in forests grow, die, and decompose, influencing their environment and their interrelationships. Forests are also disturbed. When combined, these processes and interactions define forest dynamics, and determine how much carbon accumulates at a site. None of the research domains related to forest dynamics has a complete understanding of forest carbon accumulation and variability [1–4]. Moreover, some processes and interactions are not yet understood or well described [5]. The very complexity of the forest carbon cycle is challenging for measuring, monitoring, and projecting changes to forest carbon. In addition, the geographic extent and spatiotemporal variability in the system prohibits a complete census of forest carbon [6]. As a result, no single forest carbon model, or modeling approach, currently provides a complete assessment of carbon stocks and fluxes, at any scale. Instead, multiple modeling approaches exist to quantify and understand the different processes that contribute

to forest carbon accumulation and fluxes at various spatiotemporal scales. New imperatives to commodify and account for carbon highlight the need to improve these models [7].

Regional to large-area forest carbon models range from models based on tree measurements and forest inventory data, to representations of known processes at scales varying from the leaf level to the global level. All of these models have limitations, chief among them being the need for the improved sampling of forest systems. Remote sensing can aid in alleviating some of these limitations by providing data that is spatially exhaustive, spatially explicit, and that captures change at a resolution that is commensurate with the human impact on the landscape [8]. Our objectives herein are to summarize the current state of scientific knowledge regarding forest carbon and dynamics, describe current forest carbon models and their limitations, and articulate where contributions from remote sensing science can inform next-generation forest carbon models. Our overarching goal is to expand understanding in the remote sensing community regarding the information needs that are associated with carbon modeling in general, and next-generation forest carbon models more specifically, thereby encouraging collaborative contributions to estimates of forest carbon dynamics and forest carbon modeling science.

## 2. Background: Forest Carbon Science

Forests are the largest plant community on the Earth's land surface [9], and therefore the most important terrestrial primary producers; they absorb the most carbon from the atmosphere [10]. Trees and plants convert light energy into chemical energy via photosynthesis. Stomata allow $CO_2$ into the plant, as well as water exchanges with the atmosphere, and are a central link between plants and their environment [11]. Carbon, which is stored as carbohydrate molecules, is the currency for processes such as respiration, growth, defense, and reproduction. Much plant physiology research relies on tracking carbohydrates [12]. Carbon accumulation and flux depend on the fine balance between two large central processes: photosynthesis and respiration. For example, the estimated carbon sink for the managed forests of Canada between 1990–2008, despite the large fluxes from major disturbances, was due to small differences between respiration and net primary productivity [13]. Respiration releases $CO_2$ and water back to the atmosphere at the same spatiotemporal scales as photosynthesis, making the two central processes to ecosystem structure and function [14] difficult to estimate.

Atmospheric $CO_2$ levels are rarely a limiting factor to carbon absorption; rather, temperature, radiation, and water interact to impose complex and varying limitations on vegetation productivity in different areas of the globe [15]. Figure 1 depicts the global distribution of the limiting abiotic factors to primary production in terms of water, sunlight, and temperature on a global scale. Temperature (heat) controls the rate of plant metabolism (processes), which in turn determines the amount of photosynthesis and respiration that take place. Where temperatures are between 0–50 °C [16] and water and solar radiation are available for photosynthesis, productivity is influenced by species and stand structure [17], age and site nutrients [18], the physiological adaptation of plants [19], disturbance [20], and sometimes forest management practices [21]. The level of productivity is further refined by forest phenology [22], biodiversity [23], forest stand and landscape dynamics [24], and by feedback between the changes occurring in forests and the global environment [25–28].

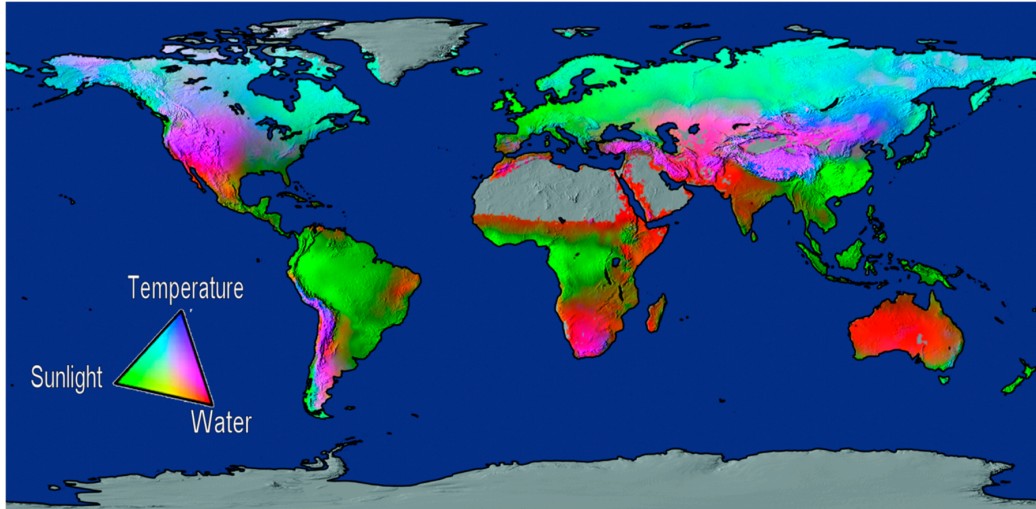

**Figure 1.** Potential limits to vegetation net primary production (NPP) based on the fundamental physiological limits of vapor pressure deficit, water balance, light, and temperature [15]. NPP is the rate of carbon accumulation in plants after losses from plant respiration and other metabolic processes (which are necessary to maintain the plant's living systems) are taken into account [29].

The overriding processes of carbon fixation (photosynthesis, respiration, and heat/water exchanges) are present at the leaf-level. Knowledge of these driving processes of carbon cycle science are relatively well established:

1.　Photosynthesis is a light-dependent process. It was described to the level of the physics of molecules in the 1980s [30], but despite this detailed description, our understanding of photosynthesis is still expanding with the advancement of quantum physics [31];
2.　Respiration is not as well defined with model choice dependent on the spatiotemporal scale under scrutiny [32–34]; and
3.　Latent heat and water interact at the leaf level via evapotranspiration and hydrological processes in forests [35,36], with research still advancing our knowledge of this complex interaction [37,38].

Moving from leaf-level processes to tree-level and forest-level processes is not additive or multiplicative [39]. The cumulative effect of fast and locally controlled processes, such as photosynthesis and respiration, combine with new processes as the spatiotemporal scale changes; trees distribute resources for the reproduction, defense, and production of fine roots, and allocate nitrogen following sometimes-undefined patterns; trees and plants change their facilitation and competition processes as they grow; forests have microbial and fungal communities that contribute to decomposition and respiration, and interact with the climate system [40,41]. Even well-defined processes can exhibit different behaviours at different scales [42,43], while other processes are still under debate [1,5,44–47]. Process interactions are extensively being studied [48–51], and entire fields of research that are determinant in forest carbon content and fluxes are evolving rapidly [1,52,53]. Research is also showing that under changing climatic conditions, species are adapting, changing our knowledge of the plasticity of tree species traits [54]. Collectively, these unknowns explain why there are currently no precise whole tree or whole forests carbon balances: we do not currently understand all the processes and their interactions [3]. Furthermore, as species are adapting to a changing environment [55], process and interactions are also changing. In short, forest carbon science is still elucidating processes and their relationships across spatial scales. In contrast to other research fields such as physics, forest ecology does not have rules or laws that transcend scales and provide consistent parameters for modeling.

## 3. Forest Carbon Models

### 3.1. The Current Carbon Modeling Continuum

The continuum of large-area forest carbon models ranges from empirical models to models of known processes that drive forest carbon balances, with all the possible combinations of these two approaches in between. While empirical models focus on describing the statistical relationships between data, process models focus on accounting for the mechanisms or processes that determine carbon accumulation in forest ecosystems. Unlike empirical models, process models are sufficiently generalized so that they remain relevant when faced with new conditions, making them suitable for future projections. Modeling approaches along this continuum vary in their level of detail (spatial, temporal, and attributional) depending on the purpose for which they were generated, the data available, and the composition and expertise of the model development team. Here, we give an overview of the different model types.

International obligations require signatory countries to report and monitor greenhouse gas balances from forests [56]. Many carbon models use the same data that is used by forest management agencies and the forest industry to estimate forest carbon for reporting purposes [57]. This includes forest inventory, land cover, land use, and ownership data, change and disturbance information, growth and yield estimates, biodiversity and wildlife management data, etc. These models have varying levels and types of ecological processes represented, with some also including simplified soil processes [58]. These carbon models range from simple statistical relationships, such as emission factors used in systems where few data are available [59,60], to complex combinations of empirical models (equations), process models (deterministic equations), and assumptions that enable these models to be executed in a timely manner [57]. It is common practice to use tree or stand-level measurements to estimate biomass using allometric equations in sampled locations, and then expand these estimates via statistical or machine learning methods to the region of interest. To obtain carbon estimates, the total biomass is then divided by a factor of two, regardless of the species or growing conditions [60,61].

Most process-based models support scientific understanding, and are often used for exploring potential responses under changing environmental conditions [62–66]. Basic sampling theory tells us that models based on observational data, such as statistical or machine learning models, are not suitable for the projection of future conditions, since the relationships and conditions that these observations represent are from the past, and these relationships and conditions are changing [67]. Hence, process-based models are a more appropriate tool for projections of future conditions. Some efforts have attempted to bridge both empirical models and process-based models [68–71]; however, these approaches are often too onerous for carbon reporting and management applications. The current limits of process understanding, including their scale-dependent behavior as well as their complicated and cumulative interactions, are the main limits of our ability to model forest carbon via process-based models. Hence, current process-based models usually represent a limited suite of well-understood processes [72].

Forests interact and influence global conditions, representing a potentially important feedback to changing climatic conditions [73,74]. However, forests are but one terrestrial biome contributing to global carbon balance estimates [75–78]. In large-area carbon models based on biogeochemical measurements [3,64,79–81], forest contributions are often calculated as the remaining balance from atmospheric and oceanic models instead of being modeled independently [82]. Atmospheric and oceanic models need to be constrained by estimates of forest carbon stock and fluxes [3,83], because forest are an integral part of the hydrological, energy, and carbon budgets in regional models [37].

## 3.2. Limitations of Current Carbon Models

Three ubiquitous limitations challenge current large-area forest carbon models: limited sample size, the population sampled relative to the population contributing to carbon storage and fluxes, and system understanding, particularly across spatial scales.

Large-area forest carbon models suffer from data paucity [84,85]. Managed forests are often the best-sampled land base in carbon modeling, wherein carbon estimates are derived from the height and diameter measurements of commercial tree species. However, those sub-populations also suffer from low sample sizes [86]. Even if we assumed that sample sizes were adequate in managed forests, these measurements are applied to allometric equations (which are also a significant source of uncertainty [87,88]) to estimate aboveground biomass [89–93], and are then divided by a factor of two to estimate carbon. Since biomass measurements are very labor-intensive, the biomass equations themselves suffer from low sample sizes, with equations developed using limited samples (i.e., in the order of hundreds of trees) and applied to species distributed across a vast geographical extent [88,92,94]. Moreover, the selection of the allometric models themselves can affect estimation [87]. Further, the 0.5 ratio to estimate carbon content from biomass estimates [60,95] also introduces another source of unaccounted error, since the carbon content of biomass, even when considering dry weight, varies much more than the 0.5 ratio that is widely used [96,97]. Process-based models, which rely on long-term measurements from sophisticated scientific equipment, typically have even lower observation data ratios, with data from a limited number of sites used to represent entire biomes [79].

Models that rely on forest inventory data for carbon estimates, such as the forest carbon balance models that are used for reporting, are generally built on a sample of a sub-population that is relevant to managed forests, such as specific commercial tree species in the productive forests that are managed for forest product extraction. Given the complexity of forest dynamics, it is unlikely that reliable carbon content and flux estimates can result from a poor sample in one strata of the forested landscape that targets specific tree species above a given diameter, even if those trees are the dominant feature of the forest biomass. Boreal forests alone, which are an ecosystem that is largely classified as unmanaged [98] and hence lacking detailed forest inventory data, are estimated to account for ~50% or more of world forest ecosystem carbon stocks, much of which is in their soils [58]. Furthermore, most allometric equations do not include biomass from other vegetation such as small diameter trees, non-commercial species, shrubs, or bryophytes, despite these components' demonstrated contributions to forest productivity [99–102]. Very few studies have quantified the carbon in non-commercial species, small-sized commercial species [103], shrubs and lichens [53,104,105], or mosses [106]. The functional and dynamic nature of all these components add to the aforementioned incomplete understanding of forests dynamics and to the challenge of modeling or predicting the productivity and carbon levels of forested ecosystems [107–109].

## 4. Potential Remote Sensing Contributions to Carbon Modeling

Remotely sensed data can aid in alleviating several of these aforementioned limitations and contribute to the next-generation of carbon modeling. Current research efforts in the remote sensing community in support of carbon modeling often assume that resulting products can be readily integrated into carbon balance assessments. However, current forest carbon models are not able to easily adapt to new data types or inputs. For example, much of the carbon-supporting research in the remote sensing community has focused on biomass estimates [110]. However, given the complexity of forest carbon estimates, their spatiotemporal variation, and the current state of forest carbon science modeling, a static estimate of biomass at a single point in time alone is inadequate information for estimating the carbon balances of large forest areas. Given our current understanding of forest carbon cycles, existing models do not provide sufficiently accurate or consistent carbon estimates across spatiotemporal scales [111]. Remote sensing science has already contributed to important improvements to large-area forest carbon modeling. However, developing the next generation of carbon models—models that are spatially explicit, that account for changes in situ, and that afford

more consistent estimates of carbon over space and time—will require a closer integration of remote sensing and carbon science in models. Research advances in remote sensing science, combined with advances in both empirical and process modeling of large-area forest carbon, can lead to improved forest carbon estimates [111]. There is an urgent need for more detailed observations to support the empirical modeling of forest carbon. Remotely sensed data can currently provide many useful inputs to both empirical and process-based forest carbon models: land cover, disturbance, leaf area index (LAI), biomass, phenology, age, height, and estimates of gross and net primary production, among others [112]. These inputs come from a myriad of sensor types: optical, radar, lidar, hyperspectral [112,113]; however, the operational readiness of these sensors to support large-area, spatially explicit modeling is not uniform [84]. The current proliferation of remote sensing missions and operational sensors at various spatiotemporal scales is poised to aid in addressing the sampling paucity and cross-scale issues that currently limit carbon modeling and system understanding. Remote sensing also seems like a logical avenue to improve the estimation and understanding of the light-driven processes at the core of carbon fixation, such as photosynthesis [114]. Better sampling supported by the remote sensing expertise to explore empirical relationships will create the opportunity to improve understanding of forest carbon cycles and their role in the large global carbon cycle and build better models. Here, we identify those limitations where remote sensing science is poised to make the greatest contribution to improving large-area carbon modeling science and application.

### 4.1. Increased Sampling in Space and Time

Remote sensing, despite not directly measuring most of the variables that are traditionally used across the spectrum of present-day carbon models, can now provide a census of forest systems [115], which is something unprecedented in forest modeling. This is the primary reason why products derived from remotely sensed data will be at the core of the next generation of forest carbon models. The variability resulting from the fast, locally controlled processes of carbon exchange (photosynthesis and respiration), layered with the increasing complexity of added processes with scale, demands that we acquire more information for modeling purposes. Remotely sensed information permits spatially-explicit modeling, capturing some of the spatial variability of these processes, which is becoming the minimum requirement for the next-generation models.

Remote sensing gives different observations than those that carbon modelers are used to: for example, energy from either the Sun or the sensor itself interacts with forest targets, and can be further interpreted to infer forest attributes [116]. These observations and their derived inferences are neither better nor worse than field data; both are surrogate measures for forest attributes, and both require due diligence and preferably cross-validation to increase confidence in their inference [117]. In both cases, forest attributes are themselves surrogates to biomass, which is a surrogate to carbon, adding two layers of modeling and their associated errors and uncertainties to the estimation of carbon. Carbon estimates from both field-based and remotely sensed observations often ignore these additional sources of uncertainty. A notable advantage of the remotely sensed information is that they can be extracted over different spatiotemporal scales and frequencies [118]. Repeated measures provide the opportunity for change measurement [116], which has been notoriously difficult to measure in forest systems, but is essential for change monitoring, system understanding, and projections of carbon fluxes.

There are vast forested landscapes that have no forest inventory, such as the northern forests of Canada [13]. Carbon estimates in these forests must rely on remotely sensed information with field validation. The few existing estimates for these areas have also combined with process modeling [119]. In these northern systems, additional challenges exist in the ambiguity of ecotones between wetlands, peatlands, and forests [120], which is another level of variability that remote sensing can help alleviate by better defining these ecotones and/or identifying their distinguishing components. To date, unmanaged forest carbon estimates are for scientific purposes, but since these northern forests are thought to contain large amounts of stored carbon [58,121], potentially released under changing

environmental conditions, it follows that complete carbon stocks and flux estimates for these areas will eventually be required for climate change policy development.

### 4.2. Upscaling of Estimates and Improved System Understanding across Spatial Scales

Increasing the frequency and extent of observations and developing inferences about forest attributes will in itself contribute to advancing our understanding of processes and their cross-scale interactions. Never before have so many observations about forests been available. Of particular usefulness is the potential that remotely sensed information offer for the scaling of various processes across spatiotemporal scales [116]. Information about forest composition, structure, productivity, and disturbances are useful for the more practical applications of carbon modeling, and are also useful for tracking changes in forests, and therefore for studying and understanding the forest system. With a proliferation of remote sensing satellites offering an increasing number and variety of observations [122], the potential for virtual constellations expands [123], further increasing the observation capacity for forests.

The rapid evolution of remotely sensed data products and methods provides at least two further opportunities: the use of machine learning algorithms, and the advancement of the cross-scale tracking of processes. In the previous state of data paucity, machine learning techniques were of little use in forest modeling. However, remote sensing provides an increasingly overwhelming observational capacity, permitting the use of these powerful tools [124], and with them, possibilities of increasing the available information and associated understanding of our forested landscapes. Although these tools do not replace statistical inference, they do increase the amount of information that is available, and can be the basis for further research that can lead to statistical inference, and eventually to more complete process modeling. The expertise for exploring the potential of these tools and their application to the plethora of remotely sensed data available lies in the remote sensing community, outside the realm of forest carbon expertise but that is essential for modeling and carbon science improvements.

One example of the process scaling that is brought by remotely sensed observations is with newly available observations of solar-induced fluorescence, which is a detectable measurement of light released during photosynthesis [125]. Photosynthesis is driven by light and is the central carbon accumulation process; it follows that observations of energy via remote sensors may have a much closer link to this driving process than tree or localized gas exchange measurements, and may possibly be even more trackable across spatial scales. This may be instrumental in our scientific understanding of the global carbon cycle, and in parallel, of forest carbon [126], since this level of productivity tracking has never been possible before.

Remotely sensed data also offer the capacity to improve the linkages between measurements acquired at different spatial scales [127,128]. Terrestrial laser scanning provides data that can improve sample sizes for allometry [129] or provide direct estimates of plot-level biomass [130]. Airborne laser scanning can provide additional data to augment ground plots [104] and link to Landsat-scale time series observations [115]. Moreover, the recent launch of two spaceborne lidars, the Advanced Topographic Laser Altimeter System (ATLAS) instrument onboard IceSat-2 [131,132] and the Global Ecosystem Dynamics Investigations (GEDI) full waveform light detection and ranging (lidar) onboard the International Space Station [133,134], provide further data for the spatial scaling of forest attributes, which is essential for carbon models [135,136]. In turn, insights enabled by cross-scale comparisons may provide further insights to modelers [137]. Exploring the potential of measuring forest or tree traits with remote sensing and modeling forest productivity also shows promise [64,138–140]. In addition to the aforementioned sensors and platforms, numerous hyperspectral satellite missions are currently in various stages of development [141], and small satellites—microsats and cubesats—have now demonstrated viability and affordability as platforms for Earth observation [142].

## 5. Discussion

Forest are complex. Our understanding of the forest system is not yet complete, and therefore, our models are likewise deficient, and our estimates are consequently variable with unknown accuracy and large uncertainties. Models, by default, push forward assumptions that are—as of yet—not supported by carbon-science findings [143]. Models often ignore scale issues for practicality, and this applies equally to those models that use remotely sensed data and those that do not.

Field data do not provide "true" carbon estimates any more than remotely sensed data do. Biases and errors need to be estimated with both data types and preferably cross validated. Declaring any exactitude in carbon estimates ignores the efforts of scientists in remote sensing science, carbon science, and the field of forest biometrics that strive to improve data products and refine models, and does not enable movement toward better science. Moreover, sensors vary in their usefulness depending on biomes [144]; biomass equations have large errors [88], tree-level biomass estimation presents many challenges [145], and errors need to be explicitly explored for different sensors [113,117,146].

There is likely no universal approach to forest carbon modeling; rather, different modeling solutions may be required in different environments, depending on the information needs, the purpose of the modeling, and the data that is available. Next-generation large-area forest carbon models will require cross-disciplinary teams to provide the appropriate specialized knowledge that is required from all of the involved disciplines. Such knowledge will enable the generation of useful remotely sensed outputs that can be incorporated into next-generation carbon models in a meaningful way [117,144]. The cross-pollination of carbon modeling and remote sensing can lead to improved model inputs, informed bias corrections [147], and more accurate estimates that account for scaling and uncertainty. Data sharing and open science is also a vital component of next-generation forest carbon models (e.g., http://forest-observation-system.net/), and indeed of all science [148].

To fully characterize the contribution of forest ecosystems to the global carbon balance, all forests need to be modeled, rather than just productive forest areas that are managed for forest products. Remote sensing science will be essential in providing data for these unproductive and unmanaged forestlands. Moreover, those components of productive forests that are not currently modeled also present particular challenges for carbon modeling, and require the support of remotely sensed observations. Further exploration of the link between the energy observations from remote sensing and the energy-based processes that drive forest productivity (light and heat) will move both fields toward better understanding and better science.

## 6. Conclusions

Carbon is the "end-product" of forest dynamics, and remote sensing is well poised for quantifying present and changing conditions in forests and therefore in carbon. The next generation of forest carbon models will include remotely sensed information, and evolve with it as well. Field data needs will remain a strong component of forest carbon modeling, since neither field nor remotely sensed observations are direct measures of carbon or carbon fluxes. However, the methods by which that field information is acquired are also evolving as a function of changing technologies. Repeated measures, both in situ and remotely sensed, are essential in this changing environment if our goal is to develop reliable models. All data is necessary to further the representation and understanding of the forest system. Models are needed that can use all levels of information, are flexible, and can adapt to new data types. That requires open science, open data, and broader collaborations. Examples of such approaches are already emerging [149]. Although forest carbon modelers are typically not prescriptive in the data sources or methods by which their model inputs are generated, they can more clearly articulate their information needs to the remote sensing community. By better understanding the information needs, the remote sensing community in turn can respond with innovative solutions and information products that are rigorously and transparently generated. Herein, we have offered some context on current forest carbon models and their challenges and limitations, highlighting the information needs of next-generation forest carbon models. From a remote sensing perspective, opportunities

to acquire observations of the globe's forests at increasingly refined spatial, spectral, and temporal resolutions have never been greater. The challenge for both remote sensing and forest carbon scientists and modelers is how to best integrate and manage these large and diverse data flows in the interest of defining and enhancing the next-generation forest carbon models, while also advancing scientific understanding of the underlying mechanisms and processes that regulate forest carbon dynamics.

**Author Contributions:** Both authors jointly conceived, researched and wrote this Technical Note in equal parts.

**Funding:** This research received no external funding.

**Conflicts of Interest:** The authors declare no conflict of interest.

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
