# Peer review of "Information Needs of Next-Generation Forest Carbon Models: Opportunities for Remote Sensing Science"

_remotesensing, doi:10.3390/rs11040463_

Round 1

Reviewer 1 Report

Boisvenue and White intended to bridge the gap between forest carbon models and remote sensing in this communication. While the authors emphasized that remote sensing could contribute inputs across spatiotemporal scales, it is unclear to the readers in the following manners:

1.       The definition of “next generation” of forest carbon model. Do you mean the process-based model that has been existed for a few decades? It is unclear from the current content.

2.       What are most important parameters in these models and how remote sensing can contribute to the mapping of these parameter? Or what do you expect the remote sensing can contribute? As the authors are experts in forest sciences, they must know the urgent needs of these forest models, while at the same time, these information gives remote sensing experts the clues they should focus on in further studies. Remote sensing has been contributing a lot, such as tree height, leaf area index, (even) biomass, phenology, burn area/ severity (and other disturbance), clumping index, et al. They also mentioned that the SIF data can contribute to model forest productivity. What’s the rest parameters (or tree traits) – and I believe that would be the value of this manuscript, while I found limited information from the current writing.

In short, I would expect that the authors can enrich the current manuscript by providing the information in needs of remote sensing researches.

Specific comments:

L67-68: I believe that the major disturbance factors should be included to get carbon sink besides R and NPP

L128: industry -> inventory?

L136: it is unclear what the coefficient 0.5 are

Author Response

Reviewer 1

Comments and Suggestions for Authors

Boisvenue and White intended to bridge the gap between forest carbon models and remote sensing in this communication. While the authors emphasized that remote sensing could contribute inputs across spatiotemporal scales, it is unclear to the readers in the following manners:

R1.1.       The definition of “next generation” of forest carbon model. Do you mean the process-based model that has been existed for a few decades? It is unclear from the current content.

RESPONSE R1.1:

We change the text to better define what we refer to as the next-generation carbon models and we changed the title of the manuscript to clarify the message conveyed by the manuscript.

Lines 228-234: However, developing the next generation of carbon models—models that are spatially explicit, that account for changes in situ, and that afford more consistent estimates of carbon over space and time—will require closer integration of remote sensing and carbon science and modellers.

Lines 2-3 Title: Information needs of next-generation forest carbon models: Opportunities for remote sensing science

R1.2.       What are most important parameters in these models and how remote sensing can contribute to the mapping of these parameter? Or what do you expect the remote sensing can contribute? As the authors are experts in forest sciences, they must know the urgent needs of these forest models, while at the same time, these information gives remote sensing experts the clues they should focus on in further studies. Remote sensing has been contributing a lot, such as tree height, leaf area index, (even) biomass, phenology, burn area/ severity (and other disturbance), clumping index, et al. They also mentioned that the SIF data can contribute to model forest productivity. What’s the rest parameters (or tree traits) – and I believe that would be the value of this manuscript, while I found limited information from the current writing.

In short, I would expect that the authors can enrich the current manuscript by providing the information in needs of remote sensing researches.

RESPONSE R1.2:

We have modified the text to clarify what is expected of remote sensing experts. Understand that our objective is not to be prescriptive on what remote sensing can provide, although we do provide plenty of examples. Rather our intent is to ensure that remote sensing scientists understand the information needs of carbon modelling extend beyond static estimates of biomass, and that—armed with this understanding, they are better equipped to see how their science and innovation can be put to best use to address these information needs.

Lines 225-248: Given current understanding of forest carbon cycles, existing models do not provide sufficiently accurate or consistent carbon estimates across spatiotemporal scales [111]. Remote sensing science has already contributed to important improvements to large-area forest carbon modelling. However, developing the next generation of carbon models—models that are spatially explicit, that account for changes in situ, and that afford more consistent estimates of carbon over space and time—will require closer integration of remote sensing and carbon science and modellers. Research advances in remote sensing science, combined with advances in both empirical and process modelling of large-area forest carbon can lead to improved forest carbon estimates [111]. There is an urgent need for more detailed observations to support empirical modelling of forest carbon. The current proliferation of remote sensing missions and operational sensors at various spatiotemporal scales is poised to aid in addressing the sampling paucity and cross-scale issues that currently limit carbon modelling and system understanding. Remote sensing also seems like a logical avenue to better estimating and understanding the light driven processes at the core of carbon fixation, photosynthesis [112]. Better sampling supported by the remote sensing expertise to explore empirical relationship will create the opportunity to improve understanding of forest carbon cycles and their role in the large global carbon cycle. Here we identify those limitations where remote sensing science is poised to make the greatest contribution to improving large-area carbon modelling science and application.

Line 301-303: The expertise for exploring the potential of these tools and their application to the plethora of remotely sensed data available lies in the remote sensing community, outside the realm of forest carbon expertise.

Line 321-326: In turn, insights enabled by cross-scale comparisons may provide further insights to modellers [135]. Exploring the potential of measuring forest or tree traits with remote sensing and modelling forest productivity also shows promise [64,136-138]. In addition to the aforementioned sensors and platforms, numerous hyperspectral satellite missions are currently in various stages of development [139], and small satellites—microsats and cubesats—now have demonstrated viability and affordability as platforms for earth observation [140].

Specific comments:

R1.3. L67-68: I believe that the major disturbance factors should be included to get carbon sink besides R and NPP

REVISION R1.3

We have changed the text to mention the contributions of major disturbances. As visible in the Figure 4 of Stinson et al. 2011 (copied below), over the 1990 to 2008 time period, fire disturbances contributed 4 TgC yr-1 to the atmosphere and transferred 27 TgC yr-1 from the live carbon pools (biomass) to the dead carbon pools. Insects accounted for a transfer of 25 TgC yr-1. Net Primary Productivity (NPP) and respiration (Rh) were an order of magnitude larger at 809 TgC yr-1 and 738 TgC yr-1 over the same time period. Based on these estimates, we maintain that the carbon sink for the managed forest of Canada between 1990-2008 was due to small differences between NPP and Rh, the two largest fluxes estimated.

Line 71-72: For example, the estimated carbon sink for the managed forests of Canada between 1990 and 2008, despite the large fluxes from majors disturbances, was due to small differences between respiration and net primary productivity [13].

R1.4. L128: industry -> inventory?

REVISION R1.4

We modified the text so that the data referred to is more specific.

Lines 139-144: Many carbon models use the same data to estimate forest carbon for reporting purposes that is used by forest management agencies and the forest industry [57]: forest inventory, land cover, land use, and ownership data, change and disturbance information, growth and yield estimates, biodiversity and wildlife management data, etcetera.

R1.5 L136: it is unclear what the coefficient 0.5 are

REVISION R1.5

We modified the text to clarify the significance of the 0.5 conversion factor between biomass and carbon.

Lines 149-154: It is common practice to use tree or stand-level measurements to estimate biomass using allometric equations in sampled locations, and then expand these estimates via statistical or machine learning methods to the region of interest. To obtain carbon, the total biomass is then divided by a factor of two—regardless of species or growing conditions - [example, 60,61].

Reviewer 2 Report

The manuscript provides a very thorough discussion on the forest carbon models and the opportunities for remote sensing in carbon modelling. The details in this manuscript are extremely useful for understanding the mechanisms of forest carbon modelling. Overall I think it is a well written paper, my only concern is that whether the focus on remote sensing is strong enough to be able to publish in Remote Sensing. I would suggest the authors either submit it to a forestry journal, or include more details on the remote sensing part.

Since the title states “Next-generation forest carbon models: Opportunities for remote sensing science”, I would expect more focus on details on the remote sensing contribution rather than traditional carbon science and modelling. For instance the authors could address more on why remote sensing data can be used to model carbon, i.e. the mechanism between forest attributes and remote sensing metrics; how remote sensing has been applied in modelling carbon and the accuracy achieved compared with traditional inventory approach (there are many studies in that topic); what sensors and approaches can be used, as well as what are some of the challenges and limitations in using remote sensing to model carbon.

Author Response

Reviewer 2

Comments and Suggestions for Authors

R2.1 The manuscript provides a very thorough discussion on the forest carbon models and the opportunities for remote sensing in carbon modelling. The details in this manuscript are extremely useful for understanding the mechanisms of forest carbon modelling. Overall I think it is a well written paper, my only concern is that whether the focus on remote sensing is strong enough to be able to publish in Remote Sensing. I would suggest the authors either submit it to a forestry journal, or include more details on the remote sensing part.

RESPONSE R2.1

We appreciate the reviewer’s comment that this manuscript is “extremely useful for understanding the mechanisms of forest carbon modelling”. We modified the text to clarify importance of this publication to the remote sensing community as—it is that community to whom this paper is targeted. As we note in the manuscript, remote sensing science has an increasingly important role to play in forest carbon modelling. By clearly articulating what the information needs of next-generation forest carbon models are, we believe the remote sensing community can better position itself to play an essential role in improving forest carbon modelling. In so doing, we have endeavoured to be concise and clear, while also providing numerous examples from the literature that support the points raised. 

Lines 225-248: Given current understanding of forest carbon cycles, existing models do not provide sufficiently accurate or consistent carbon estimates across spatiotemporal scales [111]. Remote sensing science has already contributed to important improvements to large-area forest carbon modelling. However, developing the next generation of carbon models—models that are spatially explicit, that account for changes in situ, and that afford more consistent estimates of carbon over space and time—will require closer integration of remote sensing and carbon science and modellers. Research advances in remote sensing science, combined with advances in both empirical and process modelling of large-area forest carbon can lead to improved forest carbon estimates [111]. There is an urgent need for more detailed observations to support empirical modelling of forest carbon. The current proliferation of remote sensing missions and operational sensors at various spatiotemporal scales is poised to aid in addressing the sampling paucity and cross-scale issues that currently limit carbon modelling and system understanding. Remote sensing also seems like a logical avenue to better estimating and understanding the light driven processes at the core of carbon fixation, photosynthesis [112]. Better sampling supported by the remote sensing expertise to explore empirical relationship will create the opportunity to improve understanding of forest carbon cycles and their role in the large global carbon cycle. Here we identify those limitations where remote sensing science is poised to make the greatest contribution to improving large-area carbon modelling science and application.

Line 301-303: The expertise for exploring the potential of these tools and their application to the plethora of remotely sensed data available lies in the remote sensing community, outside the realm of forest carbon expertise.

R2.2 Since the title states “Next-generation forest carbon models: Opportunities for remote sensing science”, I would expect more focus on details on the remote sensing contribution rather than traditional carbon science and modelling. For instance the authors could address more on why remote sensing data can be used to model carbon, i.e. the mechanism between forest attributes and remote sensing metrics; how remote sensing has been applied in modelling carbon and the accuracy achieved compared with traditional inventory approach (there are many studies in that topic); what sensors and approaches can be used, as well as what are some of the challenges and limitations in using remote sensing to model carbon.

RESPONSE R2.2

As per our response to the previous comments, we added text to clarify the expectation from the remote sensing community. We provide several examples of how remote sensing could (or has) contributed to forest carbon models. Our intent herein was not to extensively review the literature on the use of remote sensing to model forest carbon, but rather to highlight current deficiencies in these models that remote sensing science may be able to help address. We cite several review papers that discuss the mechanisms that link forest attributes and remote sensing metrics and describe (in detail) how remote sensing has been applied to forest carbon modelling:

1.        Goetz, S.; Dubayah, R. Advances in remote sensing technology and implications for measuring and monitoring forest carbon stocks and change. Carbon Management 2011, 2, 231-244, doi:10.4155/cmt.11.18.

2.        Lu, D.; Chen, Q.; Wang, G.; Liu, L.; Li, G.; Moran, E. A survey of remote sensing-based aboveground biomass estimation methods in forest ecosystems. International Journal of Digital Earth 2016, 9, 63-105, doi:10.1080/17538947.2014.990526.

3.        Lees, K.J.; Quaife, T.; Artz, R.R.E.; Khomik, M.; Clark, J.M. Potential for using remote sensing to estimate carbon fluxes across northern peatlands – A review. Science of The Total Environment 2018, 615, 857-874, doi:https://doi.org/10.1016/j.scitotenv.2017.09.103.

4.        Masek, J.G.; Hayes, D.J.; Joseph Hughes, M.; Healey, S.P.; Turner, D.P. The role of remote sensing in process-scaling studies of managed forest ecosystems. Forest Ecology and Management 2015, 355, 109-123, doi:http://dx.doi.org/10.1016/j.foreco.2015.05.032.

5.        Ryu, Y., Berry, J.A., Baldocchi, D.D. 2019. What is global photosynthesis? History, uncertainties and opportunities. Remote Sensing of Environment, 223: 95-114.

We posit that what the remote sensing community needs is not another review paper about how remote sensing has been used for carbon modelling, but rather a synopsis of what next generation forest carbon models need information on in order to improve estimates of carbon fluxes and system understanding.

Lines 225-248: Given current understanding of forest carbon cycles, existing models do not provide sufficiently accurate or consistent carbon estimates across spatiotemporal scales [111]. Remote sensing science has already contributed to important improvements to large-area forest carbon modelling. However, developing the next generation of carbon models—models that are spatially explicit, that account for changes in situ, and that afford more consistent estimates of carbon over space and time—will require closer integration of remote sensing and carbon science and modellers. Research advances in remote sensing science, combined with advances in both empirical and process modelling of large-area forest carbon can lead to improved forest carbon estimates [111]. There is an urgent need for more detailed observations to support empirical modelling of forest carbon. The current proliferation of remote sensing missions and operational sensors at various spatiotemporal scales is poised to aid in addressing the sampling paucity and cross-scale issues that currently limit carbon modelling and system understanding. Remote sensing also seems like a logical avenue to better estimating and understanding the light driven processes at the core of carbon fixation, photosynthesis [112]. Better sampling supported by the remote sensing expertise to explore empirical relationship will create the opportunity to improve understanding of forest carbon cycles and their role in the large global carbon cycle. Here we identify those limitations where remote sensing science is poised to make the greatest contribution to improving large-area carbon modelling science and application.

Line 322-326: Exploring the potential of measuring forest or tree traits with remote sensing and modelling forest  productivity also shows promise [64,136-138]. In addition to the aforementioned sensors and platforms, numerous hyperspectral satellite missions are currently in various stages of development [139], and small satellites—microsats and cubesats—now have demonstrated viability and affordability as platforms for earth observation [140].

Round 2

Reviewer 1 Report

Thanks to the authors to address my previous comments. Although the revision can be published at the current status, I still feel that the current content is still too abstract/general to RS readers. 

Certainly the authors are experts in the field of forest modelling. From the perspective of RS readers, they would look for what they could contribute to map specific parameters that are needs of next generation forest carbon models. They do not know so much details of the carbon cycle models, and the authors should convey such information more explicitly as the "bridge" paper. 

I made a decision of minor revision, and it is up to the authors to enrich such content. 

Author Response

RESPONSE R1.1:

Thank you to this reviewer for his/her second evaluation of our manuscript. In our previous revisions, we had modified the text to clarify what is expected/required of remote sensing experts. We re-state those changes to the text highlighted in yellow below. Our objective is not to be prescriptive on what remote sensing can provide, although we do provide numerous examples. We deliberately did not extensively review the literature on the use of remote sensing to model forest carbon, but rather highlight current deficiencies in these models that remote sensing science may be able to help address (and we identify the three most important of these needs explicitly). We believe some of the current deficiencies can be addressed with the remote sensing expertise. Our manuscript attempts to inform remote sensing scientists of where their expertise can be applied to carbon modelling beyond static estimates of biomass, and that—armed with this understanding, they are better equipped to see how their science and innovation can be put to best use to address these information needs. We believe that the way to improving our forest carbon modelling is not prescriptive but collaborative.

RESPONSE R1.2 from previous revisions:

Lines 225-248: Given current understanding of forest carbon cycles, existing models do not provide sufficiently accurate or consistent carbon estimates across spatiotemporal scales [111]. Remote sensing science has already contributed to important improvements to large-area forest carbon modelling. However, developing the next generation of carbon models—models that are spatially explicit, that account for changes in situ, and that afford more consistent estimates of carbon over space and time—will require closer integration of remote sensing and carbon science and modellers. Research advances in remote sensing science, combined with advances in both empirical and process modelling of large-area forest carbon can lead to improved forest carbon estimates [111]. There is an urgent need for more detailed observations to support empirical modelling of forest carbon. The current proliferation of remote sensing missions and operational sensors at various spatiotemporal scales is poised to aid in addressing the sampling paucity and cross-scale issues that currently limit carbon modelling and system understanding. Remote sensing also seems like a logical avenue to better estimating and understanding the light driven processes at the core of carbon fixation, photosynthesis [112]. Better sampling supported by the remote sensing expertise to explore empirical relationship will create the opportunity to improve understanding of forest carbon cycles and their role in the large global carbon cycle. Here we identify those limitations where remote sensing science is poised to make the greatest contribution to improving large-area carbon modelling science and application.

Line 301-303: The expertise for exploring the potential of these tools and their application to the plethora of remotely sensed data available lies in the remote sensing community, outside the realm of forest carbon expertise.

Line 321-326: In turn, insights enabled by cross-scale comparisons may provide further insights to modellers [135]. Exploring the potential of measuring forest or tree traits with remote sensing and modelling forest productivity also shows promise [64,136-138]. In addition to the aforementioned sensors and platforms, numerous hyperspectral satellite missions are currently in various stages of development [139], and small satellites—microsats and cubesats—now have demonstrated viability and affordability as platforms for earth observation [140].

Reviewer 2 Report

Thanks the authors for making a great effort to improve the manuscript. I appreciate your clarification of 'Our intent herein was not to extensively review the literature on the use of remote sensing to model forest carbon, but rather to highlight current deficiencies in these models that remote sensing science may be able to help address.'

Now the importance of the paper is much clearer to me and I think this paper will bring the attention of the remote sensing community to better understand how remote sensing can bring for the next-generation carbon models. 

Author Response

RESPONSE R2.1

We greatly appreciate reviewer 2’s comment as it demonstrate that we have achieved our goal in this manuscript. This reviewer clearly state that our manuscript “will bring the attention of the remote sensing community to better understand how remote sensing can bring for the next-generation carbon models.